# WHAT MAKES A GOOD PRUNE?
# MAXIMAL UNSTRUCTURED PRUNING FOR MAXIMAL COSINE SIMILARITY

**Gabryel Mason-Williams, Fredrik Dahlqvist**
Department of Computer Science
Queen Mary University
London, UK
`g.t.mason-williams@qmul.ac.uk, f.dahlqvist@qmul.ac.uk`

## ABSTRACT

Pruning is an effective method to reduce the size of deep neural network models, maintain accuracy, and, in some cases, improve the network's overall performance. However, the mechanisms underpinning pruning remain unclear. Why can different methods prune by different percentages yet achieve similar performance? Why can we not prune at the start of training? Why are some models more amenable to being pruned than others? Given a model, what is the maximum amount it can be pruned before significantly affecting the performance? This paper explores and answers these questions from the global unstructured magnitude pruning perspective with one epoch of fine-tuning. We develop the idea that cosine similarity is an effective proxy measure for functional similarity between the parent and the pruned network. We prove that the L1 pruning method is optimal when pruning by cosine similarity. We show that the higher the kurtosis of a model's parameter distribution, the more it can be pruned while maintaining performance. Finally, we present a simple method to determine the optimal amount by which a network can be L1-pruned based on its parameter distribution. The code demonstrating the method is available at `https://github.com/gmw99/what_makes_a_good_prune`

## 1 INTRODUCTION

Deep neural networks have grown in size to achieve state-of-the-art performance in various fields, e.g. object detection (Redmon et al., 2016) or natural language processing tasks (Brown et al., 2020), going from a few million parameters (He et al., 2016) to hundreds of billions of parameters (Brown et al., 2020). This increase has required substantial computational and memory resources for training and inference, making deployment difficult in resource-constrained environments.

Various approaches have been developed to compress neural networks while maintaining their performance (Blalock et al., 2020), (Gholami et al., 2021). One such approach, dating back to the 80s (LeCun et al., 1989), is neural network *pruning*, the process of removing parameters from the parent network while maintaining accuracy. Pruning methods are evaluated against each other by the percentage of parameters pruned in the network (compression ratio), along with the change in the accuracy. Pruning has been shown to be an effective method, applicable to various architectures, but reaching varying degrees of *sparsity*, the number of parameters removed from the network (Blalock et al., 2020). This leads to the following questions:

1. Why can different methods prune by different amounts yet achieve similar performance?
2. Why can we not prune at the start of training?
3. Why are some models more amenable to being pruned than others?
4. What is the maximum amount a model can be pruned before affecting the performance?

This paper aims to answer these questions and provide a deeper understanding of global unstructured magnitude pruning and neural network learning dynamics. We show that:

- Cosine similarity allows us to compare pruning methods as it is a good proxy measure for functional similarity.
- The L1 magnitude pruning method is optimal for maintaining maximal cosine similarity.
- Maintaining a high cosine similarity with the parent network ensures improved accuracy or limited drop-off in performance.
- Neural networks are *brittle*, i.e. highly sensitive to changes in high-magnitude parameters.
- The higher the kurtosis of a model's parameter distribution, the more it can be pruned while maintaining performance.
- The optimal amount by which a network can be L1-pruned can be computed explicitly from its parameter distribution. We illustrate this procedure by computing the optimal level of L1-pruning for the LeNet_Small, ResNet18 and VGG11/_BN networks.

## 2 BACKGROUND AND RELATED WORK

Neural networks are trained on a set of features, $\{x_i\}$, and corresponding labels, $\{y_i\}$, by minimizing a loss function, $L(\theta) = \frac{1}{n} \sum_{i=1}^{n} \ell(x_i y_i; \theta)$, where $n$ is the number of samples, $\theta$ represents the parameters of the network, the function $\ell(x_i y_i; \theta)$ measures how effective the network with parameters $\theta$ is at predicting the label $\{y_i\}$ for the features $\{x_i\}$ (Murphy, 2022). The number of parameters $|\theta|$ is typically very large, and the loss function defines a $|\theta|$-dimensional subspace of $\mathbb{R}^{|\theta|+1}$ referred to as the loss/objective landscape.

### 2.1 PRUNING

Pruning is defined as taking a model $\mathscr{M}(\theta)$ with parameters $\theta$ and creating a new model by applying a binary mask $M \in \{0,1\}^{|\theta|}$ to the parameters of $\mathscr{M}$ resulting in the new model $\mathscr{M}(\theta \odot M)$, where $\odot$ is the Hadamard product (Blalock et al., 2020). Thus, the pruned network operates in a parameter subspace of its parent. The two main pruning methods are structured and unstructured pruning (Blalock et al., 2020). Structured pruning removes groups of parameters, i.e. neurons and channels, whereas unstructured removes individual parameters. These methods can be applied across the whole neural network, globally, or to specific sections of the network, i.e. particular layers. The parameters are selected for pruning based on a score. The score can take many factors into account, the simplest being the magnitude of the parameters. More sophisticated factors include information from the network, such as the gradients or contributions to the network's activations (Blalock et al., 2020). After pruning, the models are fine-tuned, which commonly refers to continuing to train the network using the trained parameters of the pruned neural network. There are many pruning regimes, including *one-shot pruning*, i.e. pruning a network once and then fine-tuning, and *iterative pruning*, i.e. repeating one-shot pruning until a desired pruning percentage. A heavily pruned network will contain many zero-parameters and so be very sparse, thus sparsity measures the amount of pruning.

### 2.2 LOSS LANDSCAPES

**Recent advances.** Li et al. (2018a) have shown that many problems solved by neural networks can be solved in a smaller dimension with comparable results to the full-dimension solution and provided an upper bound for the minimum dimensions required to solve a problem. Draxler et al. (2018) and Garipov et al. (2018) have empirically shown that a path of low loss can connect two local minima. Draxler et al. (2018) proposes that these paths exist as there is a low-loss manifold that the points exist on. Garipov et al. (2018) suggests that these points are not isolated but belong to a low loss valley in the landscape. Fort et al. (2020) showed that models achieving the same loss value can represent very different functions.

**Visualisation.** Due to the high dimensionality of the loss landscape, it is intrinsically hard to visualise and reason about the dynamics of neural network learning. We will use several methods developed to reduce the dimensionality and create faithful visualisations.

First, Fort et al. (2020) showed an effective method for visualising the functional space, where the functional space is the space of all functions solving the given problem which an architecture/topology can represent. The method works by taking the model softmax outputs for a dataset,

which are flattened into a vector to represent the model predictions; the t-SNE algorithm (Van der Maaten & Hinton, 2008) is then used to reduce the dimensionality of the predictions to a two-dimensional plot, which is used to visualise the models' similarity in function space.

Second, Fort et al. (2020) also presents an effective method for visualising the loss landscape from the perspective of the origin (where all parameters are set to zero) and the two local minima points of interest. It works by perturbing the parameters radially along the weight space of two local minima points of interest $\delta, \eta$, which are then multiplied by the radial coordinates $Rx$, $Ry$. The loss of the network $L(\theta + Rx\delta + Ry\eta)$ is calculated at this point which is then plotted against $x$, $y$ to create three-dimensional representation (using colours for the third dimension) of the loss landscape. A radial slice is used as the softmax cross-entropy loss means they travel radially through the landscape during training (Fort et al., 2020).

Finally, the Filter-Wise Normalization method (Li et al., 2018b) focuses on visualising the loss landscape from the perspective of the local minima. It works by perturbing the parameters along two random approximately orthogonal filter-wise normalised directions in weight space $\delta, \eta$. The loss $L(\theta + x\delta + y\eta)$ is then plotted against the perturbation magnitudes $x$, $y$ to create an effective three-dimensional representation of the loss landscape around the local minimum. Random directions are used instead of principle component analysis (Pearson, 1901) as they allow for visualising a more expansive space instead of just the optimised parts of the landscape.

## 3   COSINE SIMILARITY PRUNING

Cosine similarity (1) is a similarity measure that measures the cosine of the angle $\varphi_{\mathbf{a},\mathbf{b}}$ between two non-zero vectors $\mathbf{a}$, $\mathbf{b}$ in an inner product space.

A similarity value of zero means the vectors are orthogonal, a value of one means the two vectors have the same direction, minus one means the vectors have opposite directions.

$$S_C(\mathbf{a}, \mathbf{b}) := \cos(\varphi_{\mathbf{a},\mathbf{b}}) = \frac{\langle \mathbf{a}, \mathbf{b} \rangle}{\|\mathbf{a}\|_2 \|\mathbf{b}\|_2} \tag{1}$$

Neural network parameters, $\theta$, are composed of several tensors of various ranks, which can be flattened by concatenation to form a vector representation of the neural network parameters. Global unstructured pruning performs a transformation on this vector $\theta$ (we use the same symbol for the network parameters and its vectorization) by zeroing some elements, thus creating a new vector $\theta'$. We will use the cosine similarity $S_C(\theta, \theta')$ to measure how much the parent network has changed under pruning. Note that the order in which tensors are concatenated when vectorizing the model does not change the cosine similarity.

The cosine similarity between the pruned sub-network and its parent will monotonically decrease as pruning increases. As a consequence, a binary search algorithm (Appendix B Algorithm 1) can be used to find the amount of pruning required to reach a target cosine similarity (within a specified $\varepsilon$).

In order to maintain the maximal cosine similarity when pruning a single parameter, one must prune the parameter with the smallest absolute magnitude.

**Theorem 1** (Maximal Cosine Similarity via Pruning)**.**

$$\max_n \frac{\langle \mathbf{a}, \mathbf{a}^{(n)} \rangle}{\|\mathbf{a}\|_2 \|\mathbf{a}^{(n)}\|_2} \text{ is reached for } n = \arg\min_i \|a_i\|. \tag{2}$$

*Where $\mathbf{a}^{(n)}$ is the vector whose components are those of $\mathbf{a}$, apart from the $n^{th}$ which is zero.*

The pruning method based on this theorem is known as the L1 pruning method since it minimises the $\ell^1$ distance. The proof of the theorem can be found in the Appendix A.

## 4   EXPERIMENTAL SETUP

To explore the questions in the introduction, we trained three distinct convolutional neural networks with varying sizes and architecture designs (Appendix C Table 1). We use the LetNet_Small

(Whitaker & Whitley, 2022) in order to have a model that is small and achieves a low performance. The ResNet18 (He et al., 2016) and VGG11_BN (Simonyan & Zisserman, 2014) networks are used as they are complex, achieve high performance, and have been shown to have different loss landscapes, with the VGG11_BN having a more complex loss landscape Li et al. (2018b).

The VGG11_BN, ResNet18 and LeNet_Small are trained on the CIFAR10 dataset (Krizhevsky, 2009), which is a dataset consisting of 10 classes with 50,000 training images and 10,000 testing images, using the Adam (Kingma & Ba, 2014) optimizer with a learning rate of 0.001 and a batch size of 256. The VGG11_BN and ResNet18 models were modified to handle the CIFAR-10 dataset as specified by Phan (2021). The VGG11_BN and ResNet18 models were trained for 100 epochs, and the LeNet_Small models were trained for 25 epochs, where an epoch is a complete pass through the training dataset.

The models were then independently globally one-shot pruned, including all the parameters, weights and biases, using Random and L1 unstructured pruning from 0 to 99% pruned with a step size of 1% and then fine-tuned for one epoch. 100% is not used as it would result in a model with zero parameters. 0% pruning is used to create a baseline of what the model would have achieved if it had been trained for the additional epoch instead of being pruned.

The ImageNet dataset (Russakovsky et al., 2015) with PyTorch-provided (Paszke et al., 2019) pre-trained VGG11, ResNet18 and MobileNetV2 (Sandler et al., 2018) models is used to demonstrate that the proposed method works at scale and highlight an edge case, see 5.4. The models were independently globally one-shot pruned and fine-tuned as described above.

## 5 RESULTS AND DISCUSSION

### 5.1 USING COSINE SIMILARITY TO COMPARE PRUNING METHODS AND OPTIMAL PRUNING

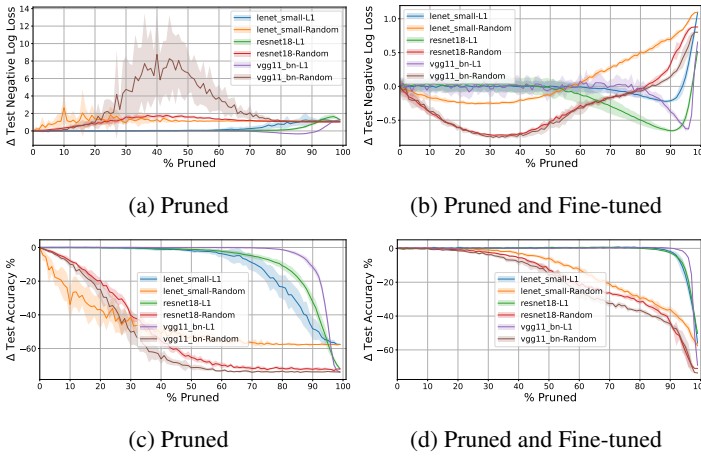

(a) Pruned

(b) Pruned and Fine-tuned

(c) Pruned

(d) Pruned and Fine-tuned

Figure 1: ReNet18, VGG11_BN and LeNet_Small change in test negative log loss and accuracy on CIFAR10 after pruning only (Figures 1a-1c) and pruning plus one epoch of fine-tuning (Figures 1b-1d) plotted against percentage pruned using Random and L1 unstructured pruning. The mean (line) and standard deviation (hue) from 10 runs are displayed.

Figure 1 illustrates the point made by the first question in the introduction that there is no relationship between, on the one hand, pruning method and pruning percentage, and on the other, changes in loss and accuracy. This can be seen very clearly in Figures 1c-1d with a marked difference in accuracy between the Random and L1 methods at 80% pruned. We see in Figure 1b that Random and L1 pruning follow similar loss trajectories: first a dip in the loss function, followed by a monotone increase as more of the network gets pruned. There is however a significant lag between the two pruning regimes, with the increase in loss occurring much earlier (at around 30% pruned) with Random pruning than with L1 pruning (at about 90% pruned). This lag is also visible in 1a if one ignores the strange behaviour of VGG11_BN under random pruning (which we cannot explain). Thus, which parameters are pruned is more important than how much of the network is pruned.

To investigate the importance of this choice we examine the effect of pruning in terms of cosine similarity with the parent network, i.e. pruning to a target cosine similarity. Figure 2 shows a much clearer relationship between the cosine similarity and the change in loss and accuracy of the network across all pruning methods. Comparing Figures 2c and 2d to Figures 1c and 1d we see that the choice of pruning method has a much smaller impact on the change in accuracy when it targets a given cosine similarity. Similarly, we see that the lag between the loss trajectories of Random and L1

pruning in Figures 1a-1b almost vanishes in Figures 2a-2b, again illustrating that cosine similarity is much more strongly associated with loss than either pruning method or sparsity/pruning percentage.

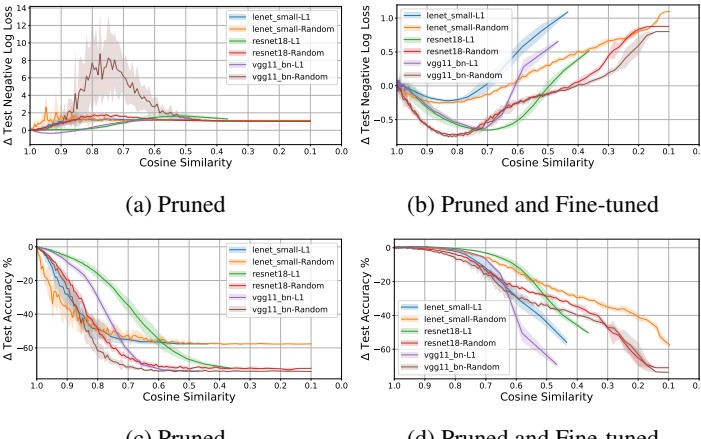

(a) Pruned          (b) Pruned and Fine-tuned

(c) Pruned          (d) Pruned and Fine-tuned

Figure 2: ReNet18, VGG11_BN and LeNet_Small change in test negative log loss and accuracy on CIFAR10 after pruning only (Figures 2a-2c) and pruning plus one epoch of fine-tuning (Figures 2b-2d) plotted against cosine similarity with the parent network using Random and L1 unstructured pruning. The mean (line) and standard deviation (hue) from 10 runs are displayed.

These observations suggest that cosine similarity is vital for understanding pruning. In particular, they suggest that an optimal pruning mechanism should maintain the highest possible cosine similarity with the parent model. Theorem 1 guarantees precisely this condition and can therefore be re-interpreted as stating that *in order to prune a vector whilst maintaining maximal cosine similarity with the original vector – the parent parameters – one must use the L1 pruning method.*

## 5.2 How Pruning Transforms the Function Space

We hypothesise that cosine similarity can often be used as an efficient proxy measure for the functional similarity between the parent and pruned network. This cannot be a universal rule. Imagine a situation where a layer $\mathcal{L}$ is connected to a network by weights of very small magnitude; this network will have high cosine similarity to the functionally very different network where these weights are pruned away, disconnecting $\mathcal{L}$ from the network. We will encounter a likely example of this phenomenon in 5.4. However, these are corner cases for which a diagnostic method will be provided. In general, we will see that cosine similarity is a very good proxy for functional similarity. If this holds, it logically follows that a network with high cosine singularity will produce a network with similar performance to the parent, as the functional representations are close. In particular, the optimal pruning method for cosine similarity (L1 pruning) should have the closest representation to the parent's function, as it removes the least impactful weights.

To explore this idea, we examine the effects of pruning on the VGG11_BN function space through the visualisation method of Fort et al. (2020). The VGG11_BN network has been shown to have a complex loss landscape (Li et al., 2018b), therefore, any effect that pruning has on the function space should be more apparent. In Figure 3, we take the VGG11_BN model's predictions at initialisation and end of training, along with predictions of the pruned models, and convert them to a function space. Figure 3 provides a functional perspective to the phenomenon illustrated through Figures 1 and 2. Figures 3a and 3b, show that the L1-pruned models stay functionally close to the parent (★) even when highly pruned, whereas the randomly pruned networks functionally diverges from the parent (very quickly in the case of Figure 3a) and, interestingly, return to a state that is functionally close to the untrained initial network (●). When pruning by cosine similarity the functional difference between L1 and Random pruning fades away. Figures 3c and 3d show that networks with similar cosine similarity with the parent are functionally close, irrespective of the pruning method. Observe from Figure 3c that for high cosine similarity values, the models remain functionally very similar to the parent, whereas models with low cosine similarity with the parent explore a region of the function space which is close to the untrained parent. After fine-tuning, Figure 3d shows that models with high cosine similarity with the parent return to a state that is functionally close to the parent but, curiously, networks with low cosine similarity with the parent again tend to return to a state functionally similar to the untrained parent model.

We hypothesise that the reason why models at a high cosine similarity can return after fine-tuning for one epoch is that they stay connected to a low loss 'valley' or 'tunnel' (Draxler et al., 2018), (Garipov et al., 2018) that leads to the local optima and, thus, can return to similar function space.

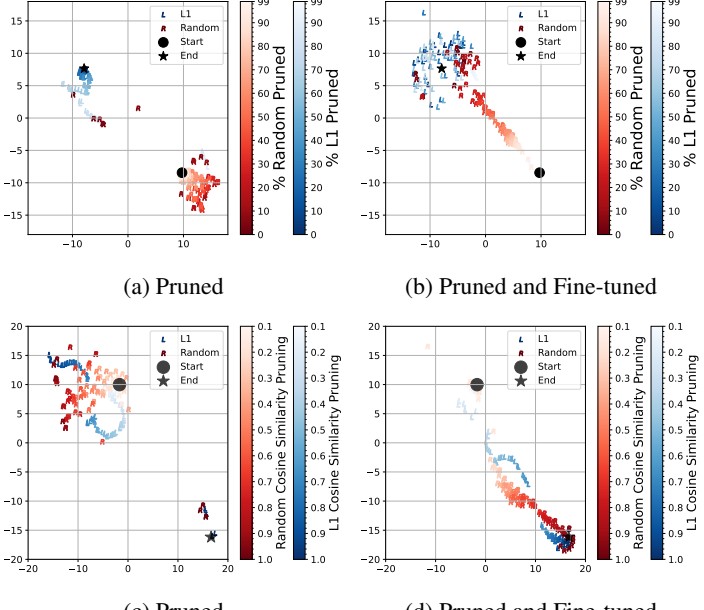

(a) Pruned             (b) Pruned and Fine-tuned

(c) Pruned             (d) Pruned and Fine-tuned

Figure 3: t-SNE projections of the training dataset predictions for the VGG11_BN model, including initialisation (●) and end of training (★). Figure 3a (resp. 3b) shows the function space of the models when pruned from 0 to 99% using Random (Red) and L1 (Blue) pruning (resp. pruning followed by one epoch of fine-tuning). Figure 3c (resp. 3d) shows the function space of the models when pruned from 1 to 0.1 cosine similarity using Random (Red) and L1 (Blue) pruning (resp. pruning followed by one epoch of fine-tuning).

We suspect that this is because the high cosine similarity forces the networks to operate in a subspace of the parent function.

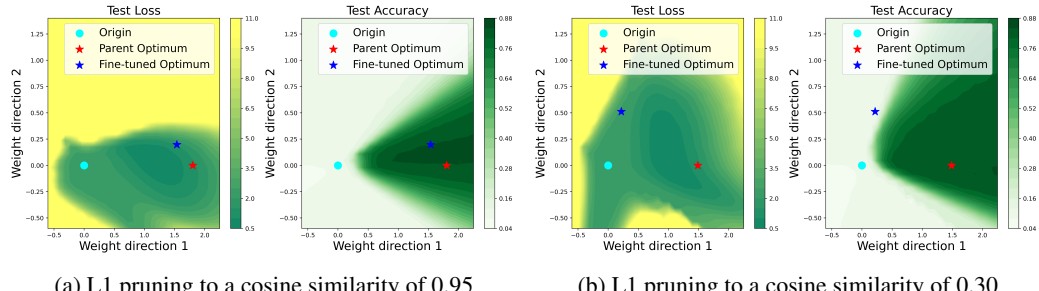

(a) L1 pruning to a cosine similarity of 0.95      (b) L1 pruning to a cosine similarity of 0.30

Figure 4: VGG11_BN Parent to Pruned-and-fine-tuned Radial Landscape slices with respect to the test data. The left-hand plots of Figures 4a-4b show the loss landscape along the paths of the parent and the pruned-and-fine-tuned model, the right-hand plots show the same landscape for accuracy.

To explore this, we follow Fort et al. (2020) and take a two-dimensional radial slice of the loss – left-hand plots in Figure 4 – and test – right-hand plots of Figure 4 – landscapes along the weight directions of the parent and the pruned-and-fine-tuned network. In Figure 4a, the network stays within the local optima 'valley'/'tunnel' for high cosine similarity but operates in a functionally different area of the loss landscape; this supports the idea of low loss manifold exists put forward by Draxler et al. (2018). For low cosine similarity, Figure 4b, the fine-tuned network becomes functionally very different from the parent. The fact that it cannot return within one epoch suggests that the network has become 'disconnected' from the low loss manifold. To see if these models can return to the low-loss region after more fine-tuning we repeated our experiments with 10 epochs of fine-tuning in Appendix D. The conclusion is that more fine-tuning amplifies what we observe with 1 epoch of fine-tuning. After a small prune, 10 epochs of fine tuning result in a network which is more similar to the parent than 1 epoch of fine-tuning. Conversely, after a big prune, 10 epochs of fine-tuning leads to a bigger functional change w.r.t. the parent than 1 epoch of fine-tuning.

## 5.3 SPARSITY, LARGE PARAMETERS AND THE LOSS LANDSCAPE

We hypothesise that neural networks tend to be (a) stable under increases in sparsity (e.g. through L1-pruning) but also (b) inherently *brittle*, that is to say they are highly impacted by changes to high-magnitude parameters. These two aspects can be quantified using the *kurtosis* of a network's weight distribution: the higher the kurtosis, the more it can be pruned (see 5.4) and the more high-magnitude

parameters it contains. It is interesting to consider the layer-wise kurtosis of a network, and in particular its *kurtosis of kurtoses* which measures the extent to which some layers have significantly more kurtosis than others. Networks with high kurtosis of kurtoses are particularly brittle: not only will they have more layers containing many large parameters, but the layer-wise kurtosis distribution will contain a peak of very low-kurtosis layers composed only of low-magnitude parameters which can get entirely pruned away, thereby disconnecting the network altogether (see 5.4).

To examine the effect of sparsity on the loss landscape, the VGG11_BN loss landscape is visualised using the Filter-Wise Normalization method (Li et al., 2018b). Figure 5b represents the loss landscape of the parent VGG11_BN network after 100 epochs of training. Figure 5a shows the loss landscape of the network after L1-pruning to a 0.99 cosine similarity. Finally Figure 5c shows the loss landscape of the network after Random-pruning to 0.99 cosine similarity. Again, we choose the VGG11_BN model because of its complex loss landscape (Li et al., 2018b).

Figure 2c showed that the pruning method has little impact on the pruned model's accuracy, the main driver of accuracy being cosine similarity with the parent. Similarly, Figure 3c showed that pruned networks with high cosine similarity stayed functionally close to their parent, irrespective of the pruning method. By examining the loss landscapes more globally in 5, we see that the pruning method does have an impact, although it is not noticeable in a small neighbourhood around the local parent network, which is why it is not apparent in Figures 2c, 3c. Indeed, comparing the loss landscapes of the two pruning methods (Figures 5a and 5c) with the loss landscape of the parent (Figure 5b), it becomes clear that:

(a) The large increase in the sparsity of the network going from parent to L1-pruned (where 70.06% of the network has been set to zero) has left the loss landscape largely unchanged ($\ell^\infty$ distance between the surfaces of 65.28).

(b) The small changes in the network caused by Random-pruning, which zeroes large parameters with the same probability as small parameters, has let to a very small increase in sparsity (only 1.99% of the parameters have been pruned) but a large change in the loss landscape dramatically ($\ell^\infty$ distance between the surfaces of 484.03).

These observations indicate that the loss landscape is heavily determined by the high-value magnitude parameters of the network. We call this sensitivity to large parameters *brittleness*.

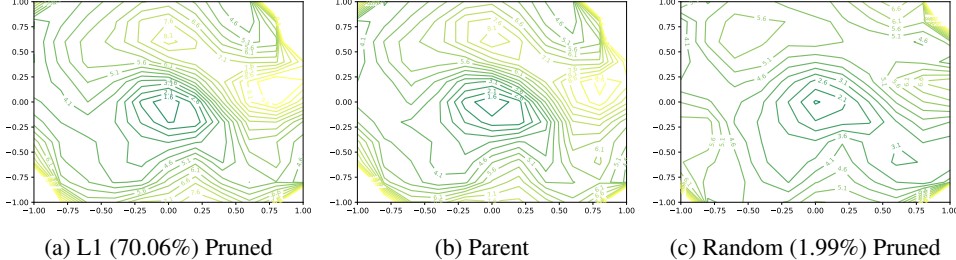

|         (a) L1 (70.06%) Pruned          |         (b) Parent          |         (c) Random (1.99%) Pruned          |

Figure 5: VGG11_BN Test Loss Landscape: (a) L1 pruning to a cosine similarity of 0.99 resulting in a 70.06% pruned network, (b) Parent network trained to 100 epochs and (c) Random pruning to a cosine similarity of 0.99 resulting in a 1.99% pruned network.

The brittleness of the models we have examined explains why a high functional similarity to the parent is required for pruning, as without this, the model does not have enough freedom to reduce the impact of the high-magnitude parameters and thus cannot easily explore other regions of the loss landscape. This idea is supported in part by Fort et al. (2020), which showed that networks with different parameter initialisations end up with different functional representations. This phenomenon is well explained by brittleness: since models are disproportionately affected by high-value parameters, and since different initialisations are very likely to assign high-values to different parameters, it is not surprising that they create such drastically different functions.

Given this observation, we suspect that high parametrisation is required early in training unless given a particularly good initialisation, as it allows for more degrees of freedom, which in turn makes it easier to transverse the loss landscape as the network is less affected by the initial initialisation of the network and parameters that would otherwise lead to poor performing model.

## 5.4 Optimal Cosine Similarity for Maximum Magnitude Pruning

When pruning is viewed through the lens of cosine similarity and, by proxy, functional similarity to their parent model, it can be reframed into a multi-objective optimisation problem, where the goal is to maximise both the cosine similarity and the percentage of the network pruned. In this framing, the L1 pruning method is the Pareto frontier, the set of all efficient solutions, and the utopia[1] would be a cosine similarity of 1 to the parent with 100% of the network pruned. The closest point to the Utopia is the optima point for pruning while maintaining the highest accuracy as it will have the maximal pruning percentage to cosine similarity ratio, i.e. either side of this point will benefit the other metric more. The exact value of the optimal point will vary depending on the distribution of the network's parameters. In particular, Figure 6 shows that the distance between the optimal pruning and utopia decreases as the kurtosis of a network's parameter distribution increases.

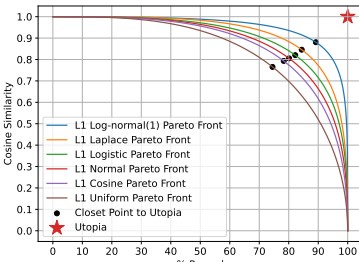

Figure 6: L1-pruned Pareto Front for 100,000 parameters distributed according to probability laws with decreasing kurtosis: Log-normal(1), Laplace, Logistic, Normal, Cosine and Uniform$[0,1]$ distributions, when optimising for Cosine Similarity to the parent and percentage Pruned. Displaying the closest point (●) on each front to the Utopia (★ at (100,1)).

This observation makes intuitive sense: a distribution with a higher kurtosis will have more high-magnitude parameters which will dominate both the numerator and the denominator of (1) in similar proportion, allowing for more low-magnitude weights to be L1-pruned whilst maintaining a high cosine similarity. It also explains why the VGG11_BN network can be pruned more than the LeNet_Small network, which can itself be pruned more than the ResNet18 network: the distributions of 10 trained networks yield average kurtoses of 8.53 for VGG11_BN, 4.82 for LeNet_Small and 3.79 for ResNet18.

The Pareto front illustrated in Figure 6 determines an optimal L1-pruning methodology. Given a trained network, the method consists in the following steps (1 and 2 are computationally cheap):

Step 1: Compute the Pareto front of the parameters (i.e. the cosine similarity from 0 to 99% pruning) and find the closest point to utopia. This gives the optimal pruning amount.

Step 2: Prune by this amount (unless the kurtosis of kurtoses is very high, in which case it might be necessary to prune by less than this amount, see MobileNetV2 example below).

Step 3: Fine-tune (by one epoch in our case).

**CIFAR10 models.** This pruning procedure is presented graphically in Figure 7 for the LeNet_Small, ResNet18 and VGG11_BN networks. Note that this procedure prunes the LeNet_Small network more aggressively than might be expected from Figure 7b, yet performance remains high after one epoch of fine-tuning (Figure 7c).

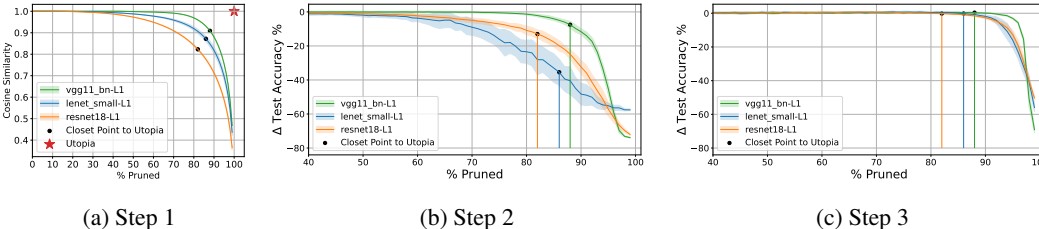

(a) Step 1        (b) Step 2        (c) Step 3

Figure 7: Optimal magnitude pruning for cosine similarity in LeNet_Small, and ResNet18 and VGG11_BN

**ImageNet models.** We demonstrate that the method described above holds at scale and with an 'out-of-sample' dataset by applying it to the VGG11, ResNet18 and MobileNetV2 models pre-trained on ImageNet and provided by PyTorch. The results are presented in Figure 8 and show something interesting. Whilst our optimal L1-pruning works well for VGG11 and ResNet18, it performs very

---

[1]By definition, the Utopia is the point whose coordinates are the optimal values of each objective. In general this point is self-contradictory and lays beyond the Pareto frontier.

poorly for MobileNetV2. The reason was described in 5.3: MobileNetV2 is an example of network containing very-low-magnitude layers which get pruned away by L1-pruning, dramatically modifying the behaviour of the network. This phenomenon can be detected by computing the parameters' kurtosis of (layer-wise) kurtoses, reporting values of 1.42 for VGG11, 5.81 for ResNet18 and 64.40 for MobileNet_V2.

The conclusion of this experiment is that for global unstructured L1-pruning, the optimal amount of pruning given by the Pareto front of Figure 6 might have to be adjusted by a parameter which decreases as the network's kurtosis of kurtoses $\kappa^{(2)}$ increases. From our limited set of experiments, if the pruning method described above produces poor accuracy, then multiplying the optimal pruning percentage by $1/\ln \kappa^{(2)}$ seems like a effective, conservative choice (see Figure 11, Appendix E). Alternatively, local (layer-wise) cosine similarity pruning might be a better strategy.

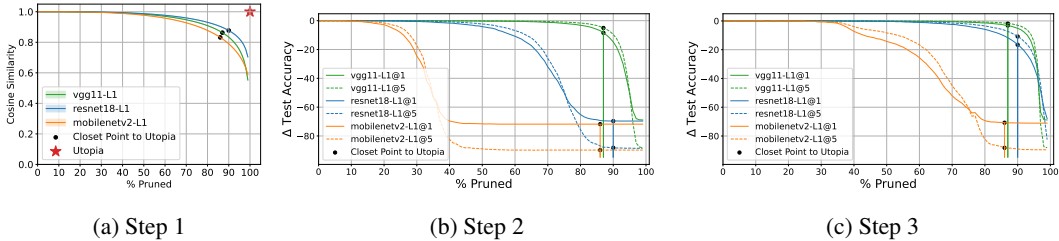

| (a) Step 1 | (b) Step 2 | (c) Step 3 |
|---|---|---|

Figure 8: Optimal magnitude pruning for cosine similarity in VGG11, and ResNet18 and MobileNetV2

## 6 CONCLUSIONS

This paper set out to quantify and explore what makes a good prune and answer the questions set out in the introduction through the exploration of global magnitude pruning.

**Why can different methods prune by different amounts yet achieve similar performance?** We showed in Section 5.1 that cosine similarity with the parent network relates pruning to performance and can be used to compare pruning methods. Two different methods, pruning by different percentages, can lead to similar cosine similarities, and thus similar performances. L1-pruning was shown to be the optimal method for maintaining cosine similarity. Moreover, we showed in Section 5.2 that cosine similarity is a good proxy for functional similarity. In particular, two different methods, pruning by different percentages, can also lead to functionally similar networks.

**Why can we not prune at the start of training?** Section 5.3 showed that neural networks are brittle, i.e. their loss landscapes can change dramatically when high-magnitude parameters are perturbed. L1-pruning a model before training would result in a model with proportionally more high-magnitude parameters (since the lower-magnitudes ones will have been pruned) and therefore highly susceptible to bad initialisation, making it very difficult to train.

**Why can some models be pruned more than others?** This is answered in Section 5.4, which shows that the optimal amount of pruning typically increases with the kurtosis of the model's parameter distribution. The exception to this rule is provided by models like MobileNetV2 on which pruning can disconnect entire layers. These situations can be detected by computing the kurtosis of layer-wise kurtoses. Models with high kurtosis of kurtoses can be pruned less that models with low kurtosis of kurtoses.

**What is the maximum amount a model can be pruned before affecting the performance?** This question is also answered in Section 5.4, which shows that the maximum amount of pruning before affecting performance is the closet point to the utopia (100% pruned with a cosine similarity of 1) on the L1 Pareto front corresponding to the parameter distribution. This point, the equal trade-off between functional similarity with the parent and sparsity of the network, can be computed exactly from the parameter distribution of the trained network, as was illustrated for the VGG11/_BN, LeNet_Small and ResNet18 networks in Figure 7 and 8.

In answering and exploring these questions, we have shown that the optimal global unstructured pruning for maximal cosine similarity is the L1 pruning method and that the optimal point for maximal pruning without dramatically affecting performance is the closet point to the utopia (100% pruned with a cosine similarity of 1) on the L1 Pareto front which can be computed efficiently.

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

## A    MAXIMAL COSINE SIMILARITY VIA PRUNING PROOF

The proof for Theorem 1 is as follows:

*Proof.* Assuming that we're working $\mathbb{R}^N$ for a fixed number of dimensions $N$. Let's write $\langle a, b \rangle$ for the usual scalar product in $\mathbb{R}^N$, namely

$$\langle a, b \rangle = \Sigma_{i=1}^N a_i b_i$$

Let's write $a^{(n)}$ for the vector whose components are those of $a$, apart from the $n^{th}$, which is set to zero. Formally

$$a^{(n)} = (a_1, \ldots, a_{n-1}, 0, a_{n+1}, \ldots, a_N)$$

We want to show that

$$\max_n \frac{\langle a, a^{(n)} \rangle}{\|a\|_2 \left\|a^{(n)}\right\|_2} \text{is reached for } n = \arg\min_i \|a_i\|. \tag{3}$$

To see that this is the case, first note that

$$
\begin{aligned}
\frac{\langle a, a^{(n)} \rangle}{\|a\|_2 \left\|a^{(n)}\right\|_2} &= \frac{\sum_{i=1} a_i a_i^{(n)}}{\|a\|_2 \sqrt{\sum_i \left(a_i^{(n)}\right)^2}} \\
&= \frac{\sum_{i \neq n} (a_i)^2}{\|a\|_2 \sqrt{\sum_{i \neq n} (a_i)^2}} \\
&= \frac{\left\|a^{(n)}\right\|_2}{\|a\|_2}
\end{aligned}
$$

Therefore,

$$\max_n \frac{\langle a, a^{(n)} \rangle}{\|a\|_2 \|a^{(n)}\|_2} = \max_n \frac{\|a^{(n)}\|_2}{\|a\|_2}$$

Since $\|a\|_2$ is fixed, the maximum is reached for the $n$ maximizing $\|a^{(n)}\|_2$. By using a Taylor expansion, we get that

$$
\begin{aligned}
\left\|a^{(n)}\right\|_2 &= \sqrt{\sum_{i \neq n}(a_i)^2} \\
&= \sqrt{\sum_{i \neq n}|a_i|^2} \\
&= \sqrt{\sum_i |a_i|^2 - |a_n|^2} \\
&= \sqrt{\sum_i |a_i|^2} - \frac{|a_n|^2}{2\sqrt{\sum_i |a_i|^2}} - \frac{|a_n|^4}{8\left(\sum_i |a_i|^2\right)^{\frac{3}{2}}} \\
&\quad - \frac{|a_n|^6}{16\left(\sum_i |a_i|^2\right)^{\frac{5}{2}}} - \dots
\end{aligned}
$$

Thus $\left\|a^{(n)}\right\|_2$ is maximized by subtracting the series in the smallest possible term $|a_n|^2$, which is precisely the smallest possible term $|a_n|$ (since squaring is monotone), in other words, the condition of equation 3. $\qquad\square$

## B  ALGORITHMS

---
**Algorithm 1** Binary Cosine Similarity Search
---
**Require:** model, similarity_to_find, epsilon
  low $\leftarrow 0$
  high $\leftarrow 100$
  **while** low $\leq$ high **do**
    mid $\leftarrow$ low $+ \frac{\text{high}-\text{low}}{2}$
    pruned_model $\leftarrow prune$(model, mid)
    similarity $\leftarrow cosine\_similarity$(model, pruned_model)
    **if** similarity $==$ similarity_to_find **then return** mid
    **else if** similarity $\geq$ similarity_to_find **then**
      low $\leftarrow$ mid + epsilon
    **else**
      high $\leftarrow$ mid - epsilon
    **end if**
  **end while**
  **return** mid
---

## C  MODEL DETAILS

| Model | No. Parameters | Batch Norm | Skip Connections |
|---|---|---|---|
| LetNet_Small | 343,402 | ✗ | ✗ |
| ResNet18 | 11,173,962 | ✓ | ✓ |
| VGG11_BN | 28,149,514 | ✓ | ✗ |

Table 1: Model Details

# D THE EFFECT OF ADDITIONAL FINE TUNING

Figure 9 shows the effect of the functional similarity when pruning by cosine similarity, and finetuning for an additional 10 epochs instead of the one epoch regieme explored in the body of the paper. It shows that after a relatively small prune, 10 epochs of fine tuning results in a network which is more similar to the original than 1 epoch of fine-tuning. Conversely, after a big prune, 10 epochs of fine-tuning will lead to a bigger functional change w.r.t. to the parent than 1 epoch of fine-tuning.

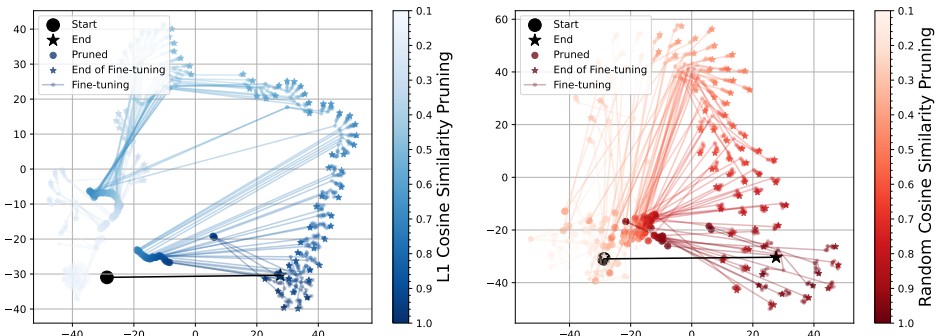

(a) L1 Pruned and Fine-tuned for 10 epochs    (b) Random Pruned and Fine-tuned for 10 epochs

Figure 9: t-SNE projections of the training dataset predictions for the VGG11_BN model, including initialisation (●) and end of training (★). Figure 9a (resp. 9b) shows the function space of the models when pruned from 1 to 0.1 cosine similarity using L1 (Blue) Random (Red) pruning followed by 10 epochs of fine-tuning.

Figure 10 shows the effect of additional epochs of training with respect to the loss landscape and the test accuracy over a range of cosine similarity values, which highlights that ten epochs of fine-tuning can yield some minor improvements, but the network essentially remains in the same loss region as one epoch of fine-tuning.

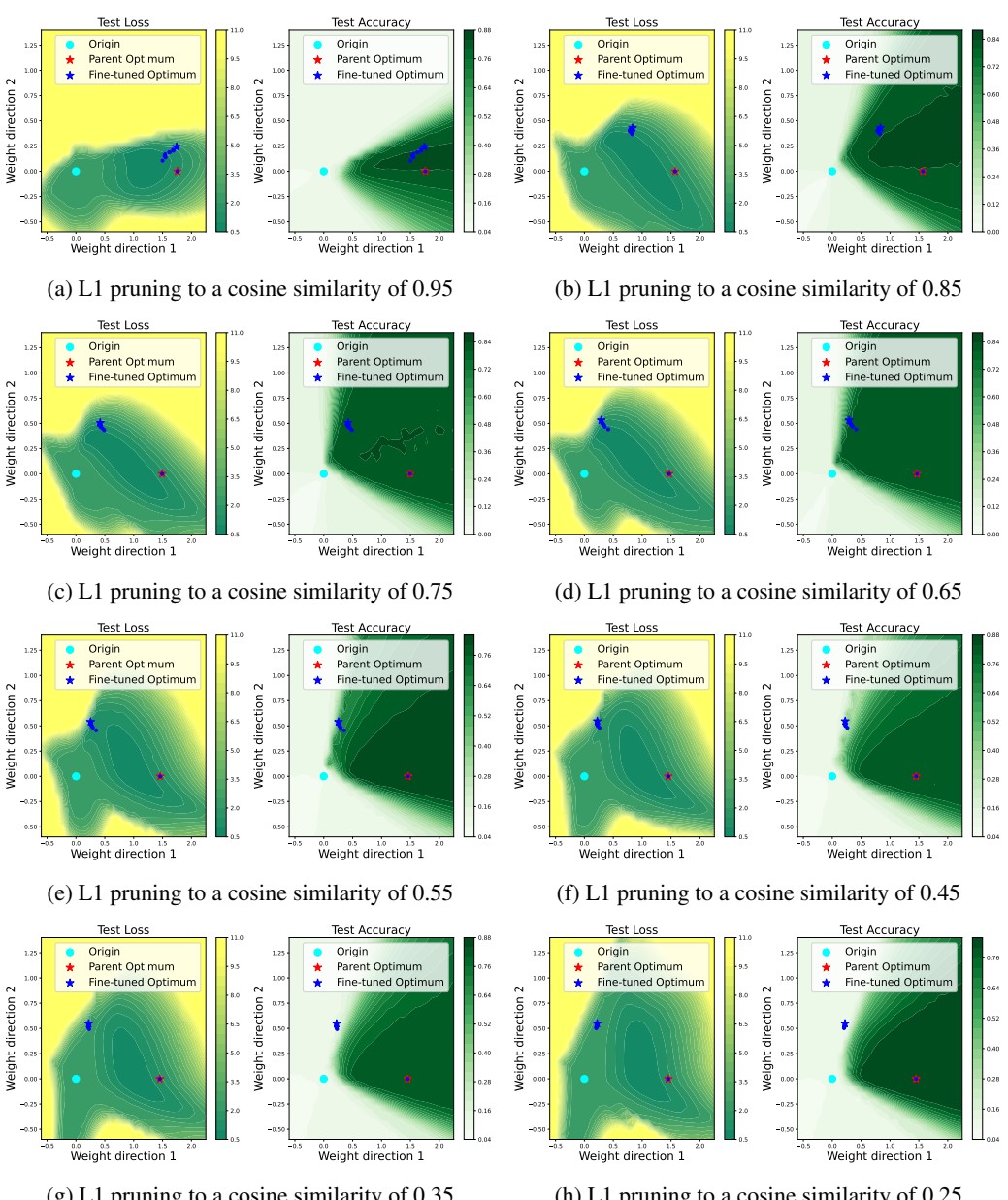

(a) L1 pruning to a cosine similarity of 0.95

(b) L1 pruning to a cosine similarity of 0.85

(c) L1 pruning to a cosine similarity of 0.75

(d) L1 pruning to a cosine similarity of 0.65

(e) L1 pruning to a cosine similarity of 0.55

(f) L1 pruning to a cosine similarity of 0.45

(g) L1 pruning to a cosine similarity of 0.35

(h) L1 pruning to a cosine similarity of 0.25

Figure 10: VGG11_BN Parent to Pruned-and-fine-tuned after 10 epochs Radial Landscape slices with respect to the test data. The left-hand plots of Figures 10a-10h show the loss landscape along the paths of the parent and the pruned-and-fine-tuned model, the right-hand plots show the same landscape for accuracy.

# E  KURTOSIS OF (LAYER-WISE) KURTOSES ADJUSTED PRUNING

In the scenario where a large kurtosis of (layer-wise) kurtoses is observed, it may be beneficial to adjust the optimal amount of pruning given by the Pareto front of Figure 6 by a parameter which decreases as the network's kurtosis of kurtoses $\kappa^{(2)}$ increases (from our limited set of experiments $1/\ln \kappa^{(2)}$ looks like a sensible, conservative choice). This is shown in Figure 11 with the conservative point shown in purple, with ResNet and MobileNetV2 having the conservative estimate of 51% and 21% respectively, given by $(optimal\_prune * 1/\ln \kappa^{(2)})$.

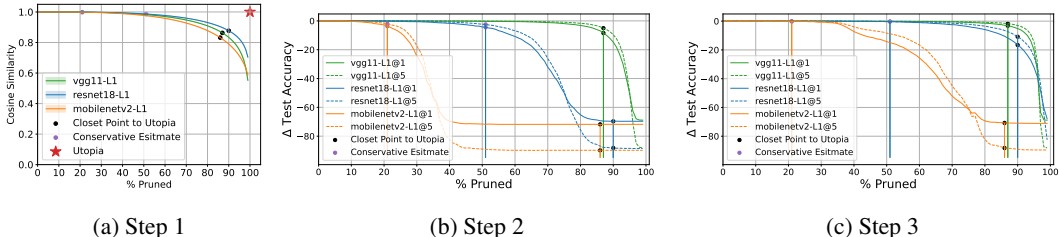

| (a) Step 1 | (b) Step 2 | (c) Step 3 |

Figure 11: Optimal and conservative magnitude pruning for cosine similarity in VGG11, and ResNet18 and MobileNetV2

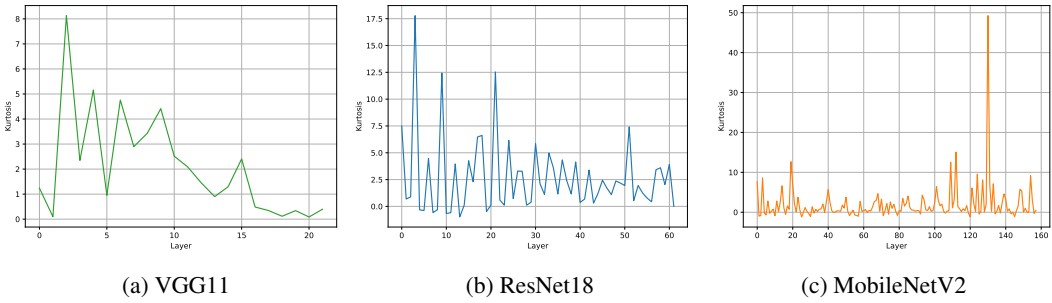

| (a) VGG11 | (b) ResNet18 | (c) MobileNetV2 |

Figure 12: Layerwise Kurtosis

