# OpenReview forum: "What Makes a Good Prune? Maximal Unstructured Pruning for Maximal Cosine Similarity"
_ICLR.cc/2024/Conference — ICLR 2024 poster_

### Official Review · Reviewer_vmPe · 2023-10-14

**Soundness:** 3 good
**Presentation:** 3 good
**Contribution:** 3 good
**Rating:** 6
**Confidence:** 4

**Summary:**

This paper presents empirical and some theoretical arguments for making maximal cosine similarity between the parent network and pruning network a more reliable target metric to focus on for optimal pruning and argues for that in a one-shot pruning regime.

The paper also presents that the longer the tail of the parameter weight distribution it is easier to prune more.

The brevity of the review doesn't stand for the quality of the review or of the paper. The paper was easy to follow and had a precise goal with only a few comments and questions from my side.

**Strengths:**

1) Motivating problem and setup
2) Precise investigation of what is important.
3) The proposal of cosine similarity as a proxy is simple, intuitive, and just works
4) The experiments help us understand that changing proxy metrics for pruning results in a more reliable way to determine better accuracy of pruned networks.
5) The empirical investigation is on CIFAR across 3 networks.
6) Furthermore investigation into loss landscapes and transformation of function space provide interesting insights into a very well-studied problem.
7) The experimentation and analysis to find the optimal cosine similarity are very interesting and further using it for pruning of neural nets to have minimal loss in accuracy.

**Weaknesses:**

1) The cosine similarity argument while intuitive and powerful is obvious from the magnitude pruning perspective -- however, what makes it interesting is the generality of it over the course of multiple 1% pruning steps.
2) I understand for every dataset network pair one can find the closest point to utopia, however, this is not sustainable, how to make this scale up across various dataset network pairs at scale?
3) My major concern is that pruning results on CIFAR-10 often are too easy and need more investigation at Tiny ImageNet and ImageNet scale to verify if the empirical insights translate. I would be very happy to increase my score and advocate for acceptance with the presence of ImageNet results on one or two networks (see Blalock et al., 2020 for best practices)

On similar lines, the networks used for CIFAR-10 are often way too overparameterized and that would be handled by experiments on ImageNet.

**Questions:**

see above

---

> ### Author Response · Authors · 2023-11-23
>
> Thank you for your constructive review.
>
> To address your main concern, we have run our pruning experiments on three networks trained on ImageNet. The results are discussed in the general comment above and detailed in the new version of the paper (specifically in sections 5.4, 5.3 and 5.2 - edits are in blue).
>
> Concerning your more specific question
>
> 2. I understand for every dataset network pair one can find the closest point to utopia, however, this is not sustainable, how to make this scale up across various dataset network pairs at scale?
>
> note that the method of finding the optimal/maximal point is CPU bound and does not require running the model on the test dataset. The approach is instead to prune the network at a resolution of 1%; this will generate 100 models at 0%, 1%, 2%,..,97%, 98%, and 99% pruned. Then, each model's cosine similarity is computed with the parent (unpruned model). This is computationally cheap and far less expensive than testing every percentage or trying to explore via a grid search for a good percentage prune.

---

### Official Review · Reviewer_AdU1 · 2023-10-27

**Soundness:** 2 fair
**Presentation:** 3 good
**Contribution:** 1 poor
**Rating:** 1
**Confidence:** 5

**Summary:**

This paper proves that the L1 pruning method is optimal when pruning by cosine similarity. Also it presents a simple method to determine the optimal amount by which a network can be L1-pruned based on its parameter distribution.

**Strengths:**

* It described the research questions that they would answer clearly early on, and then summarized them again in the conclusion.

**Weaknesses:**

*	It lacks a justification on why the consine similarity needs to be maximized. The (sub-)structure of the pruned network is different from the original network, which means their (combination of) parameters are not necessarily similar. It would have been nicer if the authors described why it should be consine-similar.

*	The paper showed that maximizing cosine similarity is to L1-prune. However, that does not tell if the L1-pruned one is anyway the optimally/best pruned network. That is because, again, similar (combination of) parameters of parent and pruned networks do not necessarily mean that the pruned network is the optimally pruned network.

*	Basically, a pruning is supposed to be retrained a lot, repeatedly. It’s unclear how valuable to show that maximizing cosine similarity is the same as removing the least magnitudes (L1 pruning), because the parameters will be retrained (fine-tuned) – then the maintained similarity will be disturbed as well. The theorem holds only when there is no retraining/fine-tuning on L1 pruning.

*	Also, the approach does 1-epoch fine-tuning. Is it just for the pruned network? Then what’s the similarity after a 1-epoch fine-tuning? Or if it does not care about cosine similarity after fine-tuning, why does it fine-tune only for 1 epoch, but not multiple times as the SOTA pruning approaches do?

*	This work lacks necessary comparisons with SOTA pruning approaches, such as Weight rewinding [1], Learning rate rewinding [2], and Gradual magnitude pruning [3][4][5]. Please consider comparing it with them. Comparing with Random pruning does not provide extremely interesting information.

*	It could be overlooked as a minor issue, but because all the results were shown with only one dataset (CIFAR10), they are not convincing. The work is encouraged to be shown with at least 3 benchmark datasets.

*	Minor typo: in page 8: in “VGG11 network can be pruned more that the LeNet Small network,”, “that” needs to be “than”.


[1] Jonathan Frankle and Michael Carbin. The lottery ticket hypothesis: Finding sparse, trainable neural networks. In International Conference on Learning Representations (ICLR), 2019

[2] Alex Renda, Jonathan Frankle, and Michael Carbin. Comparing rewinding and fine-tuning in neural network pruning. ICLR 2020

[3] Michael Zhu and Suyog Gupta. To prune, or not to prune: Exploring the efficacy of pruning for model compression. In 6th International Conference on Learning Representations, ICLR 2018, Workshop Track.

[4] Trevor Gale, Erich Elsen, and Sara Hooker. The state of sparsity in deep neural networks. CoRR, abs/1902.09574, 2019.

[5] Sidak Pal Singh and Dan Alistarh. Woodfisher: Efficient second-order approximation for neural network compression. In Hugo Larochelle, Marc’Aurelio Ranzato, Raia Hadsell, Maria-Florina Balcan, and Hsuan-Tien Lin (eds.), Advances in Neural Information Processing Systems 33: Annual Conference on Neural Information Processing Systems 2020, NeurIPS 2020.

**Questions:**

* Basically, a pruning is supposed to be retrained a lot, repeatedly. It’s unclear how valuable to show that maximizing cosine similarity is the same as removing the least magnitudes (L1 pruning), because the parameters will be retrained (fine-tuned) – then the maintained similarity will be disturbed as well. The theorem holds only when there is no retraining/fine-tuning on L1 pruning.

* The approach does 1-epoch fine-tuning. Is it just for the pruned network? Then what’s the similarity after a 1-epoch fine-tuning? Or if it does not care about cosine similarity after fine-tuning, why does it fine-tune only for 1 epoch, but not multiple times as the SOTA pruning approaches do?

* Can this work be compared with other SOTA pruning approaches?

---

> ### Author Response · Authors · 2023-11-23
>
> Thank you for your review. We'd like to make some clarifications to address your review.
>
> 1) The motivation behind this work was to (a) understand how and why global L1-pruning works, and (b) give some theoretical underpinnings to answer the question "By how much do I need to prune?". We are not trying to advocate that a particular pruning method is competitive with other methods for certain models. Comparison to SOTA is thus not entirely relevant. Rather, we are trying to understand the mechanisms of pruning. This is exemplified by e.g. our analysis of the rule of kurtosis (and kurtosis of kurtoses) in how much and how a network can be pruned.
>
> 2) Although a fine-tuning step after pruning is considered, this is not central to our argument. We are primarily concerned with pruning. Our main argument is to show that pruning by cosine similarity is a justified and effective approach to global unstructured pruning.
>
> 3) The justification is given in sec 5 where we show that:
> (a) cosine similarity is much more related to accuracy than percentage pruned (see 5.1, where random pruning is key to the argument),
> (b) cosine similarity is a good proxy for functional similarity (see 5.2),
> (c) pruning by cosine similarity does not modify the loss landscape very much (see 5.3) and
> (d) optimising cosine similarity vs percentage prune gives a computationally simple answer to the question "by how much do I need to prune?" (see 5.4).
>
> 4) What happens to the cosine similarity during fine-tuning is an orthogonal question. We know that networks become more functionally similar to the parent during fine-tuning (5.2), and it is therefore possible that the cosine similarity increases as well. Note that as a response to reviewer U3zy, we have repeated our experiments with 10 epochs of fine-tuning (see public response to this reviewer for more details).

---

### Official Review · Reviewer_cZNh · 2023-10-31

**Soundness:** 3 good
**Presentation:** 3 good
**Contribution:** 3 good
**Rating:** 8
**Confidence:** 3

**Summary:**

The paper investigates the underlying mechanisms of neural network pruning. It aims to answer questions about why different methods yield similar performance, why pruning cannot be done at the start of training, and how much a model can be pruned without affecting performance. The paper introduces the concept of cosine similarity as an effective measure for functional similarity between the parent and pruned networks. It proves that L1 magnitude pruning is optimal for maintaining maximal cosine similarity and shows that higher kurtosis in a model's parameter distribution allows for more pruning without performance loss. The paper also presents a method to determine the optimal amount of L1-pruning based on a network's parameter distribution.

**Strengths:**

1. The paper delves into the intricate mechanisms of neural network pruning, providing a understanding of why and how pruning works. This adds a layer of conceptual depth to the existing literature.

2. The paper employs rigorous mathematical proofs to substantiate its claims for the optimality of L1 pruning for maximal cosine similarity. This lends credibility to the research.

3. The paper conducts experiments on multiple architectures like LeNet Small, ResNet18, and VGG11, providing a relatively broad empirical basis for its findings.

**Weaknesses:**

1.The observation mainly made from the results on the Cifar-10 dataset, whether the observation and conclusion is extendable to other large-scale datasets remain unclear.

2. The paper focuses on specific architectures (LeNet_Small, ResNet18, and VGG11) and does not provide insights into how the findings might generalize to other types of neural networks like Transformers, or other tasks like text understanding. This contradicts the third question, which targets different models.

3. The analysis section, which comprises a significant portion of the paper, lacks logical structure and clarity.

4. Certain observations, such as the point at 80% pruned (Sect. 5.1) in Figures 1c-1d, are confusing, why 80%?

**Questions:**

See questions in weaknesses above. Additionally,

It looks to me you're computing cosine similarity of a vectorized weight vector and its pruned version, the former containing ALL weights in the network and its size would be humongous, how do you deal with that? Also, that weight vector contains weights of different DNN layers which're segregated by nonlinear activations in the network, why grouping them into one huge vector would work at all? More insights or analytical explanations are needed here.

Moreover, only LeNet, ResNet18 and VGG11 are experimented. I would be interested in seeing edge networks like MobileNet to see how "brittle" they are, and whether these edge nets are already tight for further pruning.

As mentioned, can the findings be extended to other types of neural networks, such as recurrent neural networks or Transformers?

**Details Of Ethics Concerns:**

Nil

---

> ### Author Response · Authors · 2023-11-23
>
> Thank you for your review.
>
> Your point about other types of networks is well taken. We have followed your advice and considered MobileNet, which turned out to be very informative. This model (and two others) were trained on ImageNet in order to add another training set to our experiments.  The results are discussed in the general comment above and detailed in the new version of the paper (specifically in sections 5.4, 5.3 and 5.2 - edits are in blue).
>
> Turning now to the following question.
>
> "It looks to me you're computing cosine similarity of a vectorized weight vector and its pruned version, the former containing ALL weights in the network and its size would be humongous, how do you deal with that? Also, that weight vector contains weights of different DNN layers which're segregated by nonlinear activations in the network, why grouping them into one huge vector would work at all? More insights or analytical explanations are needed here."
>
> The size, although significant, is fine as calculating the cosine similarity between two vectors can be effectively computed on the CPU and take advantage of the RAM available and thus is not limited to GPU VRAM. Similarly, computing the optimal level of pruning (see 5.4) can be done efficiently.
>
> As to grouping all of the layers into one vector, given a network topology (including choices of activation functions) and an enumeration scheme for weight matrices (e.g. stacking columns or stacking rows) and biases, there is a one-to-one correspondence between the standard representation of a network and its vectorisation. Since we're doing unstructured pruning, the network topology is fixed, which means that we can vectorise, prune the vector, and reconstruct a network without any issues.
>
> The general comment above and edits to sec 5.2-5.3-5.4 discuss what happens if a pruned segment of a vector corresponds an entire layer.

---

### Official Review · Reviewer_U3zy · 2023-11-01

**Soundness:** 3 good
**Presentation:** 3 good
**Contribution:** 2 fair
**Rating:** 5
**Confidence:** 4

**Summary:**

The authors explore the use of cosine similarity for quantifying the sparseness-accuracy tradeoff when performing unstructured global pruning of a pretrained neural network. The authors hypothesize that larger values of cosine similarity of the trained weights and the trained weights after unstructured sparsification indicate that the sparsified weights are more amenable to fine-tuning to regain accuracy lost due to pruning. To study this hypothesis, they perform an empirical study using 3 architectures of varying complexity on a single dataset and analyze the cosine similarity of two pruning strategies (random and L1). Research into this problem is motivated by the desire to better understand the complexities of model pruning (e.g., why some pruning strategies and architectures can produce sparse models with higher accuracy).

**Strengths:**

**Findings of cosine similarity on fine-tunability of pruned models:** Figures 3, 4, and 5 are interesting and summative of findings. Particularly, high cosine similarity of pruned weights to original weights enables 1 fine-tuning step to converge to point in loss landscape close to original optimum (i.e., regaining accuracy lost due to unstructured pruning).

**Weaknesses:**

**Limited evaluation:** Experiments only utilize 3 architectures, 2 pruning strategies, and 1 dataset (CIFAR10). I would expect an empirical paper at ICLR to consider at least one additional dataset (ImageNet) and some additional unstructured pruning strategies (e.g., lottery ticket rewinding) would increase impact of findings.

**Questions:**

1. At the top of p. 7 you state “It is still unclear whether, given more fine-tuning steps, these models can return to the low-loss region from their current position.” Did you consider exploring this more? I think it would be an interesting and worthwhile to empirically explore this direction by increasing the number of fine-tuning steps to see if the pruned models with lower cosine similarity can regain accuracy lost due to pruning.

2. While I find the premise and findings to be interesting, I think the evaluation is limited in that it is only performed on a single dataset. I think the addition of empirical results at least a larger scale dataset, like ImageNet, and additional unstructured pruning strategies would better support the generalizability of the takeaways.

---

> ### Author Response · Authors · 2023-11-23
>
> We have addressed your constructive criticism and Question 2 by repeating our pruning experiments on one more dataset (ImageNet) trained on several networks. The results are discussed in the general comment above and detailed in the new version of the paper (specifically in sections 5.4, 5.3 and 5.2 - edits are in blue).
>
> To answer your Question 1 we have repeated our experiments with ten epochs of fine-tuning instead of one. The conclusions (discussed in sec 5.2 and additional graphs in the appendix) are two-fold.
>
> First, additional fine-tuning magnifies the effect of the prune. After a relatively small prune, 10 epochs of fine-tuning result in a network which is more similar to the original than 1 epoch of fine-tuning. Conversely, after a big prune, 10 epochs of fine-tuning will lead to a bigger functional change w.r.t. to the parent than 1 epoch of fine-tuning.
>
> Second, in terms of accuracy, 10 epochs of fine tuning yield some small improvements but the network remains in the same loss region as 1 epoch of fine-tuning. This holds across all levels of pruning.

---

### Author Response · Authors · 2023-11-23

We thank the reviewers for their detailed reviews.

Several reviewers wanted to see our analysis of pruning applied to more and bigger models. We therefore address this point jointly in this general response.

We thank the reviewers for this suggestion. It turned out to be a very informative exercise which deepened our understanding of global pruning by cosine similarity and consequently improved the paper. We considered three models trained on ImageNet: VGG11, ResNet18 and MobileNetV2. The results and conclusions are detailed in additional comments in sec 5.2 and 5.3, and a new ending to sec 5.4. For ease of review, all our edits are coloured in blue in the revised version of the paper.

The conclusions are as follows.

1) VGG11 and ResNet18 trained on ImageNet confirms all our findings: cosine similarity remains a very good proxy for functional similarity, and pruning by optimal cosine similarity using the simple strategy described in 5.4 works very well.

2) MobileNet_V2 has an unusual architecture which highlights an interesting and inherent limitation of global L1-pruning. This network has layers consisting only of low-magnitude parameters which get pruned away almost entirely by aggressive L1-pruning. This results in severe functional changes to the network, and in particular degrades accuracy substantially.

3) The behaviour of VGG11 and ResNet18 is typical but MobileNetV2 provides an example of architecture which is not amenable to (much) global L1-pruning. We have established a simple diagnostic to detect this kind of network. Building on our observation that the kurtosis of the weight distribution relates to how much pruning can be done (5.4), we propose the kurtosis of layer-wise kurtoses as a simple measure of how much a network behaves like MobileNetV2 w.r.t. global L1-pruning. A very high kurtosis of kurtoses (as exhibited by MobileNetV2) indicates the presence of some very low-magnitude layers which are likely to be pruned away by global pruning. For such networks, less aggressive global cosine similarity pruning or local (layer-wise) cosine similarity pruning are better correct strategies.

---

### Meta-Review · Area_Chair_3Pcu · 2023-12-06

**Metareview:**

This paper studies a series of questions about unstructured pruning with cosine similarity as a guiding metric. By introducing the concept of cosine similarity, this paper proves that the L1 pruning is optimal when pruning by cosine similarity. The paper also illuminates why some models are more amenable to pruning, using kurtosis to analyze the parameter distribution within these models. Furthermore, it introduces a straightforward method for determining the optimal extent to which a network can be pruned with L1, based on its parameter distribution.
In the review process, reviewers recognize the contributions including interesting and insightful discoveries (Reviewer U3zy, cZNH, and vmPe), exhaustive evaluations (Reviewer cZNH, and vmPe), and simple yet effective method (Reviewer vmPe). However, some reviewers also express strong concerns regarding the justifications for cosine similarity (Reviewer AdU1), implementation details (Reviewer AdU1), and investigation of other pruning methods (Reviewer AdU1).
The notable divergence in opinions among the reviewers was taken into careful consideration during the decision-making process. To ensure a well-informed decision, the AC thoroughly analyzed the reviews, author responses, and the revision of the paper. The AC shares similar feelings with Reviewers U3zy, cZNH, and vmPe regarding the effectiveness of the proposed method and the insightful conclusions of this paper. Additionally, the AC acknowledged that the authors’ response effectively addresses several concerns raised by the reviewers, including experiments on more datasets and models, enhancing the quality of writing, and additional explanations of implementation details. Furthermore, there was a consensus among (part of) the reviewers favoring the acceptance of the paper. Taking all considerations into account, the AC recommends accepting this paper. However, it is advised that the authors make further revisions in the final version to address the remaining issues, particularly those highlighted by Reviewer AdU1.

**Justification For Why Not Higher Score:**

The paper did not receive a higher score primarily due to concerns raised by reviewer AdU1 about the justifications for using cosine similarity, the clarity of implementation details, and the lack of investigation into other pruning methods. These issues suggest that some parts of the paper still require further clarification and exploration to fully realize its potential.

**Justification For Why Not Lower Score:**

The paper was not assigned a lower score because the reviewers recognized several strengths, including its insightful discoveries, exhaustive evaluations, and a simple yet effective method. The authors' response in the rebuttal phase effectively addressed major concerns raised, such as experiments on more datasets and models, improving the quality of writing, and providing additional implementation details. Moreover, there was a consensus among part of the reviewers in favor of accepting the paper. These positive aspects contributed to the decision to accept the paper.

---

### Decision · Program_Chairs · 2024-01-16

Accept (poster)